# Identification and RNAi Profile of a Novel Iflavirus Infecting Senegalese *Aedes vexans arabiensis* Mosquitoes

**DOI:** 10.3390/v12040440

**Published:** 2020-04-14

**Authors:** Rhys Parry, Fanny Naccache, El Hadji Ndiaye, Gamou Fall, Ilaria Castelli, Renke Lühken, Jolyon Medlock, Benjamin Cull, Jenny C. Hesson, Fabrizio Montarsi, Anna-Bella Failloux, Alain Kohl, Esther Schnettler, Mawlouth Diallo, Sassan Asgari, Isabelle Dietrich, Stefanie C. Becker

**Affiliations:** 1Australian Infectious Diseases Research Centre, School of Biological Sciences, The University of Queensland, Brisbane, QLD 4072, Australia; r.parry@uq.edu.au (R.P.); s.asgari@uq.edu.au (S.A.); 2Institute for Parasitology, University of Veterinary Medicine Hannover, 30559 Hannover, Germany; fanny.naccache@tiho-hannover.de; 3Research Center for Emerging Infections and Zoonoses, University of Veterinary Medicine Hannover, 30559 Hannover, Germany; 4Pole de Zoologie Médicale, Institut Pasteur de Dakar, Dakar BP 220, Senegal; elhadji.ndiaye@pasteur.sn (E.H.N.); mawlouth.diallo@pasteur.sn (M.D.); 5Pole de Virologie, Unité des Arbovirus et Virus de Fièvres Hémorragiques, Institut Pasteur de Dakar, Dakar BP 220, Senegal; gamou.fall@pasteur.sn; 6Arboviruses and Insect Vectors, Department of Virology, Institut Pasteur, 75724 Paris, France; castelli.ilaria84@gmail.com (I.C.); anna-bella.failloux@pasteur.fr (A.-B.F.); 7Faculty of Mathematics, Informatics and Natural Sciences, Universiät Hamburg, 20148 Hamburg, Germany; luehken@bnitm.de (R.L.); schnettler@bnitm.de (E.S.); 8Bernhard-Nocht-Institute for Tropical Medicine, 20359 Hamburg, Germany; 9Health Protection Research Unit in Emerging and Zoonotic Infection, Public Health England, Porton Down, Salisbury SP4 0JG, UK; jolyon.medlock@phe.gov.uk; 10Medical Entomology & Zoonoses Ecology, Emergency Response Department Science & Technology, Public Health England, Porton Down, Salisbury SP4 0JG, UK; ben.cull@phe.gov.uk or; 11Department of Medical Biochemistry and Microbiology/Zoonosis Science Center, Uppsala University, 75237 Uppsala, Sweden; jenny.hesson@imbim.uu.se; 12Laboratory of Parasitology, Istituto Zooprofilattico Sperimentale delle Venezie, 35020 Legnaro (Padua), Italy; fmontarsi@izsvenezie.it; 13MRC-University of Glasgow Centre for Virus Research, Glasgow G61 1QH, UK; alain.kohl@glasgow.ac.uk; 14German Centre for Infection Research, partner site Hamburg-Lübeck-Borstel-Riems, 20359 Hamburg, Germany; 15The Pirbright Institute, Pirbright GU24 0NF, UK

**Keywords:** *Aedes vexans*, iflavirus, RNAi, virus discovery

## Abstract

The inland floodwater mosquito *Aedes vexans* (Meigen, 1830) is a competent vector of numerous arthropod-borne viruses such as Rift Valley fever virus (*Phenuiviridae*) and Zika virus (*Flaviviridae*). *Aedes vexans* spp. have widespread Afrotropical distribution and are common European cosmopolitan mosquitoes. We examined the virome of *Ae. vexans arabiensis* samples from Barkédji village, Senegal, with small RNA sequencing, bioinformatic analysis, and RT-PCR screening. We identified a novel 9494 nt iflavirus (*Picornaviridae*) designated here as Aedes vexans iflavirus (AvIFV). Annotation of the AvIFV genome reveals a 2782 amino acid polyprotein with iflavirus protein domain architecture and typical iflavirus 5’ internal ribosomal entry site and 3’ poly-A tail. Aedes vexans iflavirus is most closely related to a partial virus sequence from *Venturia canescens* (a parasitoid wasp) with 56.77% pairwise amino acid identity. Analysis of AvIFV-derived small RNAs suggests that AvIFV is targeted by the exogenous RNA interference pathway but not the PIWI-interacting RNA response, as ~60% of AvIFV reads corresponded to 21 nt Dicer-2 virus-derived small RNAs and the 24–29 nt AvIFV read population did not exhibit a “ping-pong” signature. The RT-PCR screens of archival and current (circa 2011–2020) *Ae. vexans arabiensis* laboratory samples and wild-caught mosquitoes from Barkédji suggest that AvIFV is ubiquitous in these mosquitoes. Further, we screened wild-caught European *Ae. vexans* samples from Germany, the United Kingdom, Italy, and Sweden, all of which tested negative for AvIFV RNA. This report provides insight into the diversity of commensal *Aedes* viruses and the host RNAi response towards iflaviruses.

## 1. Introduction

The floodwater mosquito *Aedes vexans* (Meigen, 1830) (Diptera: Culicidae) are hematophagous insects found worldwide and distinguished as three subspecies based on adult morphological traits [1]. *Aedes vexans vexans* is the most common subspecies in Europe, Canada, the United States of America, parts of East Asia, and also the Australasian and Oceanic Islands [2]. *Aedes vexans nipponii* (Theobald) is found in China, Korea, and Japan [1]. The African subspecies *Aedes vexans arabiensis* (Patton) is found abundantly in the Arabian Peninsula, Ethiopia, and Sudan and can be distinguished morphologically as having broader pale tergal bands and shorter male palps [3].

*Aedes vexans* spp. are competent vectors of arthropod-borne viruses (arboviruses) such as Rift Valley fever virus (RVFV) (*Phenuiviridae*) [4], Tahyna virus (*Peribunyaviridae*) [5], Zika virus, West Nile virus (ZIKV and WNV, *Flaviviridae*) [6,7], Getah virus [8], and Eastern equine encephalitis virus (*Togaviridae*) [9]. In addition to arboviruses, *Ae. vexans* are also implicated in the transmission cycle of the filarial nematodes *Dirofilaria immitis* and *Dirofilaria repens* [10].

Metagenomic studies examining the ecology and diversity of viruses associated with *Aedes* mosquitoes have shown that, in addition to arboviruses, *Aedes* mosquitoes host several commensal insect-specific viruses (ISVs) [11,12]. Insect-specific viruses are notable due to the fact of their insect restricted host range, and experimental infection of whole mosquitoes and cell lines suggest that ISVs can restrict or enhance arbovirus transmission [13,14,15,16]. Metagenomic analyses of *Ae. vexans nipponii* pools collected in the Republic of Korea has resulted in the characterization of four insect-related viruses: Yongsan bunyavirus 1 (YBV1) (*Bunyaviridae*), Yongsan picorna-like virus 3 (YPLV3) (unclassified *Picornavirales*), Yongsan sobemo-like virus 1 (YSLV1) (related to arthropod *Solemoviridae*) [17], and the insect-specific flavivirus Chaoyang virus [18]. Except for studies on *Ae. vexans nipponii*, little is known about the commensal virome of other *Ae. vexans* spp.

In *Aedes* mosquitoes, three main classes of small RNAs respond to and modulate virus infection. These are microRNAs (miRNAs), small-interfering RNAs (siRNAs), and P-element-induced wimpy testis in *Drosophila* (PIWI)-interacting RNAs (piRNAs) (reviewed in References [19,20,21]). During virus infection, the riboendonuclease III enzyme Dicer-2 processes double-stranded RNA (dsRNA) produced during viral replication into virus-derived short interfering RNAs (vsiRNAs) 20–25 nt in length with a strong bias for 21 nt in dipterans [22]. The vsiRNAs are then loaded into the RNA-induced silencing complex (RISC), where they target RNA molecules through complementarity and reduce virus accumulation and promote tolerance in the host [23]. As small-RNA samples in arthropods are enriched for virus-derived small RNAs, it is possible to *de novo* assemble virus genomes from small RNA sequencing [24].

In this study, we examined small RNA data from two laboratory colonies of *Ae. vexans* for the presence of viruses in these samples. We report the *de novo* assembly and annotation of a novel iflavirus, provisionally named Aedes vexans iflavirus (AvIFV), and examine the siRNA and piRNA responses targeting AvIFV in the host. Iflaviruses are non-enveloped positive-sense RNA viruses belonging to the order *Picornavirales* with a monopartite genome 9–11 kilobases in length [25]. Currently, one mosquito iflavirus has been formally isolated and characterized, Armigeres iflavirus infecting *Armigeres* spp. mosquitoes in the Philippines [26]. Iflavirus genomes encode a single open reading frame flanked by 5’ and 3’ untranslated regions (UTR) with the 3’ UTR containing a poly (A) tail [25]. The *Iflaviridae* family gets their namesake from infectious flacherie virus, which can cause premature mortality in the silkworm, *Bombyx mori* (Lepidoptera: Bombycidae). However, not all iflaviruses cause pathology with many isolated from asymptomatic insect hosts.

To examine the incidence and natural ecology of AvIFV infection in *Ae. vexans* spp., we performed RT-PCR screening of Senegalese and European mosquito pools. AvIFV is present in both archival and current laboratory colonies as well as in wild-caught *Ae. vexans arabiensis* mosquitoes from Barkédji village, Senegal. Intriguingly, we were unable to identify AvIFV reads in high-throughput sequencing from *Ae. vexans vexans* laboratory colony samples obtained from Germany and maintained at the Institut Pasteur Paris (France) or in RT-PCR screens of field-collected specimens of other European countries (Germany, the United Kingdom, Italy, and Sweden). The results suggest that AvIFV is unlikely to be a widespread commensal virus of European *Ae. vexans* mosquitoes, but also that AvIFV is ubiquitous and asymptomatic in Senegalese *Ae. vexans arabiensis* mosquito hosts.

## 2. Materials and Methods

### 2.1. Mosquito Samples and Collections

For the virus discovery presented in this manuscript, we used two different RNA seq datasets produced in the framework of previous projects. The first dataset originates from *Ae. vexans arabiensis* mosquito samples used for small RNA sequencing approach which analyzed the production of small RNA after Rift Valley fever virus MP-12 strain infection [27]. For the original infection experiment, the F_1_ mosquitoes from a lab colony established from a ground pool in Barkédji village (15°17′ N, 14°53′ W) were used. Female mosquitoes were blood-fed with rabbit erythrocytes containing RVFV (MP-12 strain) and eleven infected female mosquitoes pooled at fifteen days post blood meal [27]. The second dataset used for the presented study derived from *Ae. vexans vexans* colony mosquitoes from Kühlkopf, Germany, (49°49′ N, 8°27′ E [28]). For sequencing, mosquitoes were blood-fed with rabbit erythrocytes (control) or blood-fed with rabbit erythrocytes containing RVFV (MP-12 strain) for 48 h and pooled. For RT-PCR screens of Senegalese *Ae. vexans arabiensis*, four groups of mosquitoes were collected and tested: Aevex1—adult F_1_ generation of *Ae. vexans arabiensis* individuals collected in 2013 used for vector competence studies triturated in Leibovitz-15 (L-15) medium (GibcoBRL, Grand Island, NY, USA) with 20% fetal bovine serum (FBS; Gibco BRL, Grand Island, NY, USA); Aevex2—F_1_ generation of a lab colony of *Ae. vexans arabiensis* set-up in 2020 triturated in Glasgow Minimum Essential Medium (GMEM, Thermofisher Scientific, Waltham, MA, USA); Aevex3—*Ae. vexans arabiensis* collected at adult stage in the field at Barkédji village in 2011 and triturated in GMEM; and Aevex4—*Ae. vexans arabiensis* from a lab colony of the same F_1_ generation as Aevex1 used for infection studies triturated in GMEM. For RT-PCR screens of European *Ae. vexans vexans*, adults were collected in 2017 in Germany (*n* = 171), 2019 in Sweden (*n* = 25), 2019 in Italy (*n* = 10) as well as *Ae. vexans vexans* collected in 2018 from Nottinghamshire (United Kingdom) (*n* = 86) (refer to Table 1 for detailed locations).

### 2.2. RNA Extraction and Small RNA Deep Sequencing

For *Ae. vexans arabiensis* samples, RNA extraction and sequencing were undertaken as previously described [27]. Raw fastq files were downloaded from the National Center for Biotechnology Information (NCBI) Sequence Read Archive (SRA) under the accession: SRR5457466. Sequencing from *Ae. vexans vexans* samples made available for this study were carried out by Edinburgh Genomics (University of Edinburgh) using an Illumina HiSeq 2000 platform. Raw fastq data for *Ae. vexans vexans* samples were deposited in the SRA under the accession PRJNA565664.

### 2.3. Bioinformatics: Virus Discovery and RNAi Profile Analysis

The adapters of the small RNA-Seq libraries were identified and trimmed using Trim Galore (Galaxy Version 0.4.3.1) with reads shorter than 15 nt discarded. Clean reads were subsequently assembled using SPAdes (3.11.1) with *k*-mer values (23, 25) using the –rna flag [29]. For virus discovery, assembled contigs were queried against a local non-redundant virus database using Basic Local Alignment Search Tool (BLASTx) [30]. This database consists of all deposited virus proteins from Genbank collapsed for redundant sequences using CD-HIT [31]. For assembly of AvIFV, several contigs ranging between 912 nt and 8590 nt were identified by BLASTx as related to the polyprotein of Wuhan insect virus 13 [32]. The putative Wuhan insect virus 13 related fragments were subjected to reciprocal BLASTn to identify overlapping regions. Overlapping regions were identified and stitched together and overlapping regions of the genome as well as additional validation with the assemblers Trinity (v2.1.1) and CLC Genomics Workbench (12.0.3) (Qiagen, Aarhus, Denmark) can be seen in Appendix A. The stitched AvIFV genome was validated through re-mapping to the AvIFV genome using Bowtie2 (Galaxy Version 2.3.4.3) [33] under default conditions. Coverage and mapping statistics were determined with the bedtools (v2.27.1) genome coverage tool [34] and visualized using Microsoft Excel 2016. For analysis of the host RNA interference (RNAi) response to the virus, reads mapping to the AvIFV genome were extracted from the binary alignment file using BAM filter (Galaxy version 0.5.9). For visualization of vsiRNA (21 nt) and vpiRNA (24–29 nt) mapping profiles, reads were trimmed to individual nucleotide lengths and subsequently remapped to the AvIFV genome using Bowtie2 as previously described. Nucleotide biases in 27 nt vpiRNAs were extracted and visualized using WebLogo [35]. The assembled AvIFV genome was deposited to Genbank under the accession number MN314969.

### 2.4. Virus Genome Annotation and Phylogenetic Analysis

Open reading frames (ORFs) of AvIFV were predicted using the NCBI Open Reading Frame Finder (https://www.ncbi.nlm.nih.gov/orffinder/) with a minimum ORF length of 100 aa using the standard genetic code. The predicted polyprotein was analyzed for domains using the Conserved Domain Search Service (https://www.ncbi.nlm.nih.gov/Structure/cdd/wrpsb.cgi). Putative transmembrane domains were identified and discriminated from signal peptides using the TOPCONS web server (http://topcons.cbr.su.se/pred/) [36]. The internal ribosomal entry site was annotated using IRESPred web server (http://bioinfo.net.in/IRESPred/) [37]. For phylogenetic placement of AvIFV within *Picornaviridae*, we used the Bayesian Markov chain Monte Carlo (BMCMC) coalescent framework using BEAST (v2.5.1) [38]. Closely related iflaviruses were determined through querying the predicted polyprotein of AvIFV against the non-redundant protein database. Additionally, we queried the AvIFV polyprotein against the Transcriptome Shotgun Assembly (TSA) Sequence Database on NCBI using tBLASTn to identify putatively assembled, but un-annotated viruses. In total, 35 representative polyprotein sequences as well as representative *Picornaviridae* outgroup viruses were downloaded from NCBI and aligned using MUSCLE [39]. Uninformative or ambiguously aligned regions were removed using the GBlocks web server 0.91 (http://molevol.cmima.csic.es/castresana/Gblocks_server.html) [40] and subjected to phylogenetic analysis with BEAST under a Yule model with the following conditions: chain length: 10 × 10^6^, sampling every 10,000 steps, Whelan and Goldman (WAG) amino acid substitution model. Convergence for all parameters was inspected using Tracer v1.7.1 (effective sample sizes > 1000). The maximum clade credibility (MCC) tree was selected from the posterior tree distribution using the TreeAnnotator program, considering a 10% burn-in.

### 2.5. Construction and Validation of an AvIFV RT-PCR Screen for *Ae. vexans spp.* Mosquitoes

Mosquito samples previously described were divided up into pools of up to six adults and screened as homogenates for AvIFV using an RT-PCR strategy. *Ae. vexans arabiensis* samples from Senegal were split into forty pools for testing. *Ae. vexans vexans* samples used for testing were from Germany (*n* = 34 pools), United Kingdom (*n* = 13 pools), Italy (*n* = 5 pools), and Sweden (*n*= 2 pools) (Table 1). Mosquitoes were individually homogenized in 500 µL GMEM using the automated sample homogenization machine TissueLyser II (Qiagen, Hilden, Germany) including one 5 mm steel bead (Isometall, Pleidelsheim, Germany) and crushed under following conditions: 30 s, 30 oscillation/s. The suspensions were clarified by centrifugation (5000× *g* for 1 min), and the supernatant was used for RNA extraction. The RNA extraction was conducted on crushed mosquito supernatants using QIAamp Viral RNA Kit (Qiagen, Hilden, Germany) according to the manufacturer’s protocols. To screen *Ae. vexans* mosquitoes for potential infection with AvIFV, an RT-PCR-based detection method was established using specific primers amplifying AvIFV gRNA at positions 8713–8983 (Fw: 5’GACAGATGCGTTGAAGAGTGGT’3 and Rv: 3’TGCCCGTTTAATAGCTTCGCA’5). The reactions were performed in a 12.5 µL reaction volume using Superscript^TM^ III one-step RT-PCR Kit (Invitrogen, Karlsruhe, Germany) on a peqStar 96X thermocycler (Peqlab, Erlangen, Germany) amplifying a 271 bp product, according to the manufacturer’s instructions. As an additional DNA control, a 294 bp fragment of the AvIFV genome was synthesized and cloned into the pEX-A128 plasmid backbone (Eurofins, Germany) and used in tandem in PCRs. An internal RNA extraction control (eGFP 10^8^ c/mL) was included in the protocol testing every pool using RT-PCR [41]. The mosquito housekeeping gene (actin-1) was used as a second control to monitor the integrity of mosquito sample RNA according to previously published protocols [42]. The thermal cycler profile was as follows: reverse transcription, and initial amplification was performed at 60 °C for 1 min and 50 °C for 45 min followed by denaturation at 94 °C for 2 min, and 45 cycles of 94 °C for 15 s, primer specific annealing temperature of 57 °C for 30 s and 68 °C for 40 s, and a final extension step at 68 °C for 7 min.

## 3. Results

### 3.1. Aedes vexans arabiensis Mosquitoes from Barkédji Village, Senegal, Were infected with a Novel Iflavirus

We analyzed small RNA sequencing data from female adult *Ae. vexans arabiensis* mosquitoes (Barkédji village, Senegal) and *Ae. vexans vexans* colony samples maintained at Institut Pasteur Paris, France. Both colonies had been experimentally infected with RVFV in a recent examination of the influence of the RNAi response in Culicinae mosquitoes [27]. Excluding assembled RVFV contigs, only the *Ae. vexans arabiensis* library had contigs bearing similarity to any other viral proteins through BLASTx analysis (E-value < 10^−5^) to a local virus database. A number of putative virus fragments (the largest 8590 nt) were most closely related to the polyprotein of Wuhan insect virus 13 (GenbankID: YP_009342321.1), assembled from a metagenomic study of pooled insects in China [32]. Extension of the 5’ and 3’ ends identified using reciprocal BLASTn analysis, manually stitched and positioned in a single orientation. The genome assembly strategy and benchmarking using different assemblers are available in Appendix A. The virus genome was then validated through remapping to the draft genome resulting in the virus genome 9494 nt in length with an average coverage of 4691 and a minimum read depth of nine at every position.

Prediction of the genes of this putative virus revealed a single large (8349 nt) ORF. The BLASTp analysis of the predicted polyprotein against the non-redundant virus database indicated that the most closely related virus in pairwise protein identity is the partial *Venturia canescens* picorna-like virus polyprotein genome (GenbankID: AAS37668.1, Identity: 56.77%, Query cover: 17%) identified in the endoparasitic wasp *Venturia canescens* (Hymenoptera: Ichneumonidae) [43]. The most closely related complete genome was Wuhan insect virus 13 (GenbankID: YP_009342321.1, Identity: 38.84%, Query cover: 93%) [32]. Other closely related relatives identified through BLASTp analysis are members of the *Iflavivirdae,* and as such we provisionally assigned the name Aedes vexans iflavirus (AvIFV) to this contig. The nucleotide composition of AvIFV is A/U rich (60.6%) which is similar to that of other iflaviruses such as deformed wing virus (DWV) (61.26%) [44], Varroa destructor virus (VDV) (61.41%) [45], and infectious flacherie virus (IFV) (57.2%) [46].

The 5’UTR region of iflaviruses contains an IRES which promotes the translation initiation of the polypeptide in a cap-independent manner. Analysis of the 718 nt 5’UTR of AvIFV using the IRESPred webserver estimated it was likely to encode an IRES based on homology and the presence of multiple stable RNA hairpin structures and one Y-shaped RNA structure (IRESPred Output: Appendix A).

The polyprotein of iflaviruses is highly conserved and consists of a short leader protein (L) that is removed from the capsid protein VP2 before the assembly of the structural capsid proteins, arranged in the order VP2–VP4–VP3–VP1. The AvIFV polyprotein contains several conserved picornavirus structural domains: a cricket paralysis virus capsid protein-like domain (Pfam accession: PF08762, E-value: 5.3 × 10^−9^) between residues 981 and 1126, a calicivirus coat protein between residues 590 and 726 (Pfam accession: PF00915, E-value: 0.00058), and a picornavirus capsid protein (Pfam accession: PF00073, E-value: 0.56) between residues 979 and 1081 corresponding to VP3 and VP1. During iflavirus capsid assembly, structural precursor proteins are autocatalytically cleaved by the virus 3C-protease [47]. We identified a 3C cysteine protease domain through alignment between residues 2188 and 2210 with the conserved GxCG and GxHxxG motifs [47] (Appendix A) as well as likely conserved 3C-protease substrate sites were predicted between structural proteins (Figure 1). The non-structural regions of AvIFV were also highly similar to iflaviruses with the conserved RNA helicase predicted between residues 1464 and 1570 (Pfam accession: PF00910, E-value: 1.01 × 10^−10^). The RNA-dependent RNA polymerase domain (Conserved Domain accession: CD01699, E-value: 1.67 × 10^−54^) was identified at the C-terminal end of the polyprotein between residues 2419 and 2712 (Appendix A). Additionally, while the iflaviruses DWV [44], VDV [45], and IFV [46] are known to encode a 2A-like motif -DxExNPGP- [48], we were unable to identify any complete or partial 2A-like regions in the AvIFV polyprotein.

For phylogenetic placement of AvIFV within *Iflaviridae*, we queried the polyprotein sequence of AvIFV against the NCBI non-redundant database using BLASTp and downloaded closely related viruses based on similarity. We also identified several likely iflaviruses from assembled transcriptomes deposited on the transcriptome sequence archive (TSA) using the tBLASTn algorithm under default parameters. We aligned the polyprotein from AvIFV, related iflaviruses, and predicted polyprotein sequences from TSA along with members of the *Districtoviridae* and *Picornaviridae.* The resultant alignment was subjected to Bayesian evolutionary analysis sampling trees (BEAST) analysis, and the maximum clade credibility tree (Figure 2) placed AvIFV with high posterior probability to a clade encompassing Wuhan insect virus 13 and a likely iflavirus identified from the transcriptome of *Bombus pyrosoma* (Apidae). In the phylogenetic analysis presented here iflaviruses do not cluster together with related insect hosts, and, notably, AvIFV is distantly related to Armigeres iflavirus (ArIFV). The distant genetic relationship of AvIFV and ArIFV suggests that iflaviruses infecting mosquitoes appear to have arisen through distant evolutionary histories. This genetic distance could be explained by either poor sampling of the viruses between these clades or potentially host switching may have occurred within this family.

### 3.2. The Virus-Derived Small RNA Profile of Aedes Vexans Iflavirus

Previously, the RNAi response against the iflavirus DWV in honeybees (*Apis mellifera*) was examined [49,50]. Similarly, we sought to investigate the composition and mapping profile of vsiRNAs against AvIFV in *Ae. vexans* mosquitoes. In total, 155,124,463 adapter trimmed reads representing the small RNA fraction (18–32 nt) were mapped against the AvIFV genome using Bowtie2. Approximately, 1.23% of the entire library (1,913,033 reads) mapped to both the genome and the anti-genome of AvIFV.

Generally, the profile of small RNA reads indicated a slight bias towards the genome orientation and that an overwhelming majority of AvIFV reads (~60%) were 21 nt in length (Figure 3A). Reads representing this 21 nt peak were then extracted and re-mapped to the genome. Visual inspection of the mapped 21 nt vsiRNA population suggests total coverage of the AvIFV genome as well as equal distribution along the AvIFV genome (coverage $\bar{x}$ = 1227x) and the anti-genome (coverage $\bar{x}$ = 1227x) strands with no “hot spots” observed (Figure 3B). The mapping profile and 21 nt bias suggest that these vsiRNAs are produced by Dicer-2-mediated cleavage of the dsRNAs produced during virus replication and importantly suggest that AvIFV is actively replicating in this mosquito. In addition to vsiRNAs, we examined the virus-derived RNA population of the length 24–29 nt which correspond to vpiRNAs. Historically, piRNAs were thought to silence transposons at the transcriptional level. However, the presence and production of vpiRNAs in *Aedes* spp. have been experimentally determined (e.g., [51]). Reads between 24 and 29 nt were mapped to the genome and visually inspected for hot spots. In comparison to vsiRNAs, it appears the most mapped 24–29 nt AvIFV reads originate from the genome orientation of AvIFV (Figure 3C). Furthermore, these 24–29 nt reads were examined for a piRNA ping-pong signature with antisense piRNAs having a 1U bias, sense piRNAs having a 10A bias, and overlap between sense and antisense piRNAs of 10 nt. While these mosquitoes have been demonstrated to produce vpiRNAs against RVFV [27], we were unable to identify any such piRNA characteristics in the AvIFV 24–29 nt reads (Figure 3D).

### 3.3. RT-PCR Screening of Ae. vexans spp. Samples from Senegal, Italy, the United Kingdom, Germany, and Sweden Suggests AvIFV is Exclusively Present in Ae. vexans Samples from Senegal

To estimate the prevalence of AvIFV in *Ae. vexans arabiensis* samples from Senegal and *Ae. vexans vexans* adults from European samples, we designed an RT-PCR primer assay to amplify a 271 bp fragment of the AvIFV genome from pools of mosquitoes. In total, we screened homogenates from forty *Ae. vexans arabiensis* pools from Barkédji village Senegal and 54 pools collected from seven German sites, one British site, five Italian sites, and one Swedish site (Table 1). All *Ae. vexans arabiensis* mosquito samples from Senegal tested positive for AvIFV. These samples originated from females wild collected in 2011 (Aevex3 pools), the F_1_ generation lab colony samples established in 2013 (Aevex1 pools and Aevex4 pools) as well as a current 2020 lab colony that originated from a second independent lab colony (Aevex2). These data suggest that *Ae. vexans arabiensis* mosquitoes are not only ubiquitously infected by AvIFV in the wild, but also that AvIFV can persist for several years in natural *Ae. vexans* populations.

For European *Ae. vexans vexans* samples, we were unable to detect any AvIFV genomic RNA (gRNA) using the same AvIFV primer set. Since a positive sample from AvIFV-infected *Ae. vexans arabiensis* mosquitoes were not available in our European laboratory at the time of the study, we established two DNA controls: firstly, a commercially synthesized gene fragment containing a 294 bp AvIFV gDNA and the same AvIFV gDNA with T7 promoter sequence in a Topo TA vector backbone. These gDNA controls were later validated using positive RNA from *Ae. vexans arabiensis* mosquito samples by our laboratory in Senegal. Both positive RNA from *Ae. vexans arabiensis* mosquito samples and our gDNA controls led to the corresponding amplicon using AvIFV primers in RT-PCR and PCR under the same conditions the (Appendix A); therefore, we considered this an adequate control for the screening of European mosquitoes. Additionally, false-negative results were excluded through the use of a GFP internal control for RNA extraction as well as *Ae. vexans actin-1* to ensure mosquito RNA integrity (Appendix A).

## 4. Discussion

With the increasing use of high-throughput sequencing and the improvement of bioinformatics tools to facilitate the discovery of viral genomes, the detection of insect-specific viruses (ISVs) has steadily increased. Amongst those ISVs, iflaviruses constitute a rapidly growing family. Initially described in lepidopterans *B. mori* [46], *Spodoptera exigua* [52], and *Antheraea pernyi* [53], and hymenopterans such as *Vespula pensylvanica* [54] and *A. mellifera* [44], iflaviruses are mostly asymptomatic in their insect host [25].

While iflaviruses have currently been described in different mosquito species including *Armigeres* [26], *Culex, Mansonia* [55], and *Aedes* (this study), all three of these mosquito iflaviruses show limited relatedness to one another.

As stated above, pathological changes attributed to iflavirus infection are rare and thus far only described for DWV [44], IFV [46], and Antheraea pernyi iflavirus (ApIV) in which ApIV is thought to be the causative agent of *A. pernyi* vomit disease [53]. In line with this, we did not observe apparent disease symptoms in our F_1_ colony from Senegal, or other wild-caught mosquitoes sampled over a 9-year window suggesting that AvIFV is likely to be a covert infection. The presence of numerous unannotated iflavirus-like sequences in assembled transcriptomes from other arthropods supports the general covert nature of many iflavirus infections [26]. In contrast to the F_1_ colony from Senegal, samples of the *Ae. vexans vexans* colony from Germany (Institut Pasteur colony) as well as all field-collected *Ae. vexans vexans* samples from Europe were negative for AvIFV. These results suggest a very low incidence, if any presence at all, of this virus in *Ae. vexans vexans* population from Europe. Other studies analyzing the prevalence of iflaviruses in natural insect populations showed a very low prevalence for Culex picorna-like virus (CuPV-1) with minimum infection rates of 2% (7/340) in *Culex* and *Mansonia* mosquitoes from Mozambique [55] and low infection rates of 13.1% for SeIV-1 and 7.7% for SeIV-2 infection in *S. exigua* populations from Spain [56]. In contrast, 33 out of 36 apiaries throughout France (100 bees for each apery) tested positive for DWV, indicating a very high prevalence of this virus in bee populations [57]. However, such tests can be biased due to the fact that iflavirus infection seems to spread rapidly in colonies once infected individuals are introduced [56]. This might lead to a very high incidence of DWV in bee colonies due to the exchange of infected queens among the apiaries. All in all, most iflaviruses show low prevalence in natural populations, and it might well be that we missed natural infection due to the restricted set of samples analyzed.

Currently, we can only speculate how AvIFV might be transmitted between *Aedes* mosquitoes. In this report, samples from two independent colonies established from wild-caught eggs tested positive for AvIFV, indicating that vertical transmission was likely to have occurred in these samples. Vertical transmission of a number of iflaviruses has been experimentally determined and observed in lepidopteran and hymenopteran hosts. For example, SeIV1 and SeIV2 transmit from infected mothers to offspring in both experimentally infected *S. exigua* colonies [52] and in natural populations [56]. ApIV was transmitted vertically by 20% of the experimentally infected individuals and naturally infected field-collected breeding couples [53]. In addition to vertical transmission, vector-borne transmission of DWV by *Varroa* mites has also been demonstrated [25]. Finally, oral transmission of iflaviruses has been suggested as a potential route of infection [53]. However, this has not been shown empirically in the iflaviruses mentioned above [53].

It is of interest that AvIFV was identified in samples experimentally infected with RVFV [27]. This leads to the question of how the double infection might influence infection parameters of either virus. Thus far, reports on ISVs and dual-host arbovirus co-infection in mosquitoes are conflicting. For example, Culex flavivirus (CxFV) infection suppresses WNV infection in *Cx. pipiens* [58], and Palm Creek virus suppresses WNV and Murray Valley encephalitis virus (*Flavivirus*) infection in mosquito cells [59]. In contrast, either no effect on WNV propagation or a slight enhancement of transmission was observed in *Cx. quinquefasciatus* co-infected with CxFV Izabal [60] and CxFV did not impact RVFV infection in *Cx. pipiens* mosquitoes [61]. Co-infection studies using cell-fusing agent virus (CFAV, *Flavivirus*) and dengue virus (DENV) in Aag2 cells shows mutual interaction through activation of Ribonuclease K (AeRNASEK) expression [14], whereas in *Ae. aegypti* mosquitoes, CFAV negatively interfered with DENV-1 and ZIKV infection [15]. These results demonstrate that interaction of ISVs and arboviruses are highly system specific. Further, virus interactions at the whole organism level may produce a different outcome as compared to the cellular level. In lepidopterans, co-infections of iflaviruses and the large dsDNA baculovirus (*Baculoviridae*) show either decreased [62] or enhanced [63] pathogenicity of the baculovirus, but in both studies, the environmental stability and infectivity of the iflavirus were increased. Whether ISVs, such as an iflavirus, interact with dual-host arboviruses in hematophagous insects is not clear due to the lack of available data. In light of the increasing evidence of iflavirus infections in mosquitoes, such studies would be of vital importance to improve the understanding of arbovirus-mosquito interactions.

While the discoveries of ISVs in vector mosquitoes are ever-increasing, few reports have characterized the interaction of ISVs with the host immune system to further understand potential arbovirus infection suppression phenotypes [64]. Here, we were able to explore the interaction of AvIFV with the antiviral RNA interference pathway through the examination of the abundance and distribution of vsiRNAs and vpiRNAs derived from AvIFV sequences. The most abundant virus-derived small RNAs appear to be the 21 nt AvIFV-derived vsiRNAs. These correspond to Dicer-2 cleavage products, and further examination of these 21 nt fractions showed an equal distribution of reads along the genome targeting both viral genomic and anti-genomic sequences and thus indicating efficient targeting of AvIFV replication intermediates by *Ae. vexans* Dicer-2 protein.

The RNAi response against both RVFV and AvIFV can be ascertained in this library, and the absolute abundance of AvIFV-derived vsiRNAs exceeded those for RVFV-derived vsiRNA (data presented in Reference [27]). We assume that either the targeting of AvIFV dsRNA replication intermediates is more efficient by the Dicer-2 protein or that the dsRNA replication intermediates are more abundant for AvIFV compared to RVFV. The latter might be because positive-strand RNA viruses may produce more Dicer-2 accessible dsRNA than negative-strand RNA viruses such as RVFV [65].

Interestingly, we were not able to detect any AvIFV-derived 24–30 nt RNAs with the piRNA signature in our dataset. However, high numbers of RVFV-derived piRNAs could be identified within the same dataset [27]. This indicates that (1) *Ae. vexans* are capable of piRNA production, and (2) no technical issues are preventing piRNA detection in those samples. Thus, the absence of AvIFV-derived piRNAs can only be explained by a lack of targeting by the piRNA machinery, at least by mechanisms resulting in piRNAs with the expected signature. Several other studies analyzing the targeting of viruses by siRNA and piRNA pathways in mosquitoes have shown that while the siRNA pathway targets all viruses, only some virus families are prone to produce ping-pong based piRNAs. Generally, negative-sense RNA bunyaviruses [66], mononegaviruses [13,67], and the single-stranded DNA densoviruses [68] produce 24–30 nt vpiRNAs. In addition, all tested arboviruses belonging to the alphavirus genus produce ping-pong based piRNAs in contrast to the insect-specific Agua salud alphavirus [69]. For other viruses, viral specific pi-like small RNAs have been reported either solely due to the fact of their size range of 24–30 nts or intermediate vpiRNA signatures (i.e., A_10_, but not U_1_ bias [70]). The available data, taken together, suggests that individual virus families, be it dual-host arboviruses or ISVs, are more often targeted by the piRNA machinery than others at the current state of knowledge. If there is any link between the piRNA production and control of these viruses in the germline and, for example, vertical transmission, is not clear.

## Figures and Tables

**Figure 1 viruses-12-00440-f001:**
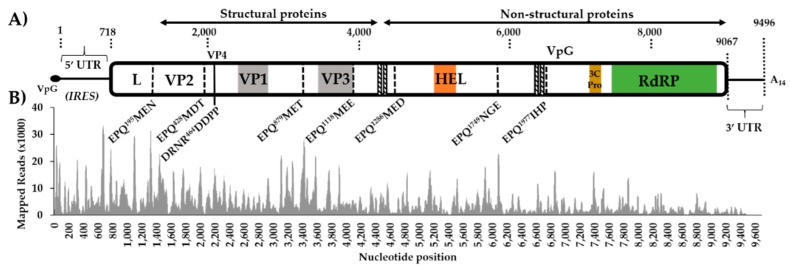
Genome organization of Aedes vexans iflavirus (AvIFV) and mapping coverage. (**A**) Genome organization schematic of AvIFV with dotted lines indicating conserved 3C-protease cleavage sites inferred by amino acid alignment and conserved protease consensus sites. The VP1 region corresponds to a calicivirus coat domain (PF00915), and VP3 corresponds to Cricket paralysis virus capsid (PF08762) domain. HEL corresponds to predicted RNA helicase domain (PF00910), 3C-Pro corresponds to 3C/3C-like protease domain, and RdRP corresponds to the RNA-dependent RNA polymerase domain (PF00680). Diagonal striped lines indicate predicted transmembrane domains. internal ribosome entry site (IRES) of 5’UTR (Appendix A). (**B**) Mapping coverage of AvIFV genome 18–44 nt trimmed reads re-mapped to both genome and anti-genome orientation of the AvIFV genome.

**Figure 2 viruses-12-00440-f002:**
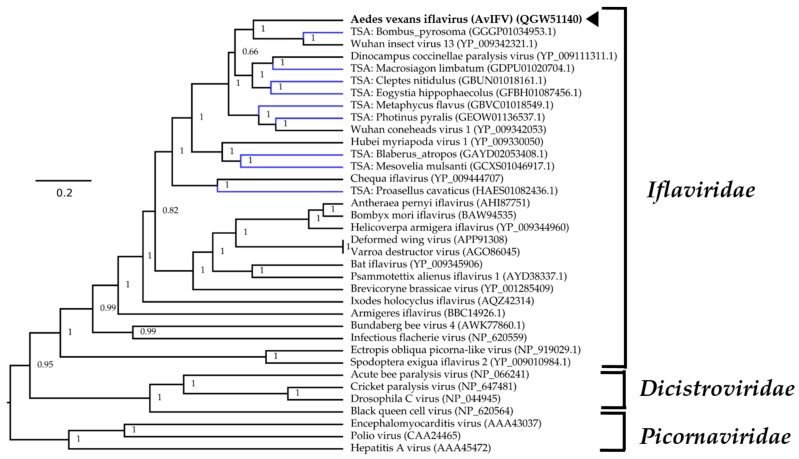
Maximum clade credibility tree of AvIFV within the *Picornavirales* order. Phylogenetic inferences were calculated using Bayesian evolutionary analysis sampling trees (BEAST v2.5.1) with polyprotein alignments using the Whelan and Goldman protein substitution model with a chain length of 10 × 10^6^. A black arrowhead indicates AvIFV and Genbank accession number is in parentheses. Putative viruses identified in the Transcriptome Shotgun Assembly are labelled in blue. The tree is rooted arbitrarily on the *Picornaviridae* outgroup. Branch length indicates the number of substitutions per site.

**Figure 3 viruses-12-00440-f003:**
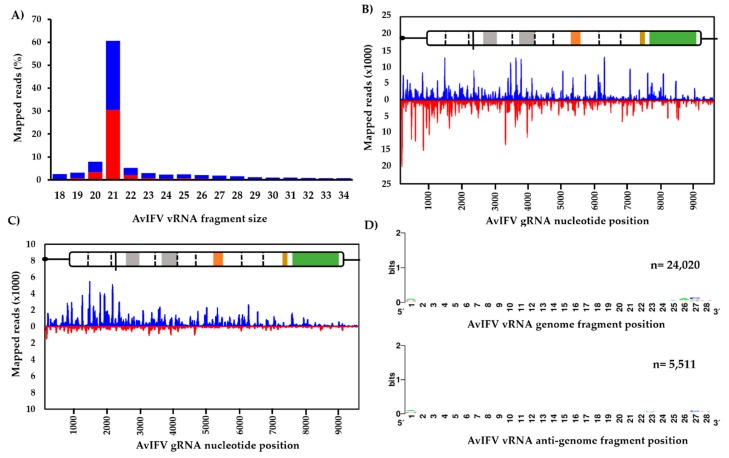
Virus-derived small RNAs derived from AvIFV replication in adult *Ae. vexans arabiensis* mosquitoes. (**A**) Profile of nucleotide lengths of virus-derived small RNAs mapped to the AvlFV genome. (**B**) Profile of 21 nt virus-derived short interfering RNAs and (**C**) virus-derived piRNA reads (24–29 nt) mapped to the AvIFV genome (blue) and anti-genome (red). (**D**) AvIFV does not produce vpiRNAs with the characteristic piRNA signature (adenine at position 10, A_10_, for sense RNA; upper panel and Uridine at position 1, U_1_ for antisense RNA; lower panel).

**Table 1 viruses-12-00440-t001:** Table listing the field or laboratory origins of *Ae. vexans* spp samples as well as the collection site, collection date, the number of pools screened (1–6 mosquitoes per pool), and Aedes vexans iflavirus RT-PCR positive (+), or negative (–) result. Countries: Senegal (SEN); Germany (GER); the United Kingdom (UK); Italy (ITA); Sweden (SWE). Calendar months: MAR (March); JUN (June); JUL (July); AUG (August); SEP (September).

Country	Location	Laboratory or Field	Coordinates	Date	Pools Screened	RT-PCR Result
SEN	Barkédji village	Laboratory	15°17′ N, 14°53′ W	MAR 2020	10	+
SEN	Barkédji village	Laboratory	15°17′ N, 14°53′ W	2013	10	+
SEN	Barkédji village	Laboratory	15°17′ N, 14°53′ W	2013	10	+
SEN	Barkédji village	Field	15°17′ N, 14°53′ W	2011	10	+
GER	Großheide	Field	53°36′ N, 7°21′ E	JUL 2017	1	−
GER	Varel	Field	53°24 N, 8°8′ E	JUL 2017	1	−
GER	Rastede	Field	53°14′ N, 8°12′ E	JUL 2017	1	−
GER	Oldenburg	Field	53°6′ N, 8°15′ E	JUL 2017	1	−
GER	Bremen	Field	53°5′ N, 8°51′ E	AUG 2017	1	−
GER	Hamburg	Field	53°28′ N, 9°49′ E	AUG 2017	12	−
GER	Frankfurt a.M.	Field	50°6′ N, 8°40′ E	AUG 2017	3	−
GER	Frankfurt a.M.	Field	50°6′ N, 8°40′ E	SEP 2017	14	−
UK	Nottinghamshire	Field	53°7′ N, 0°54′ W	JUL 2018	13	−
ITA	Guarda Veneta	Field	44°58′ N, 11°47′ E	JUN 2019	1	−
ITA	Porto Tolle	Field	44°54N, 12°27′ E	JUN 2019	1	−
ITA	Papozze	Field	44°59′ N, 12°2′ E	JUN 2019	1	−
ITA	Ceggia	Field	45°40′ N, 12°39′ E	JUN 2019	1	−
ITA	Riese Pio X	Field	45°43′ N, 11°54′ E	JUN 2019	1	−
SWE	Forshaga	Field	59°31′ N, 13°29′ E	JUN 2019	2	−

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
