# Peer review of "Identification and RNAi Profile of a Novel Iflavirus Infecting Senegalese Aedes vexans arabiensis Mosquitoes"

_viruses, 2020, doi:10.3390/v12040440_

Round 1
Reviewer 1 Report
The authors have adequately addressed the concerns raised in my previous review.
Author Response
We thank the reviewer for his nice comment. We are happy that he likes the new manuscript.
Reviewer 2 Report
The authors report the discovery of an insect-specific virus using high-throughput genomic sequencing technologies. Using bioinformatic techniques, they were able to assemble a genome, which resembled known iflaviruses. They describe the putative genome and report a small relative abundance in laboratory and wild-caught mosquito samples from Europe and Africa.
The manuscript is very well-written, the methods and study design are clear, and the results are presented nicely.
This topic is fascinating, but remains rather niche. I think future studies of ISVs would greatly improve our understanding of insect - specifically mosquito - immunity, which would in turn greatly improve our understanding of vector competence. Therefore this manuscript - combining both virus discovery as well as characterizing the immune response - will surely provide fundamental literature for the future research. In that sense, it is a shame that the virus, itself, was not isolated, as the material seems readily available and the experiment would be straight forward (using insect cell lines and in vivo infection of naive European mosquitoes).
I have very few critical comments/suggestions - and really these are just minor suggestions. The manuscript is ready for publication.
1. The authors state that Aedes vexans is an invasive species, and I'm not sure if there are data to support that. It is certainly widespread ("cosmopolitan") species and locally quite abundant, but I am not sure whether it could be classified as invasive.
2. Please mention in the opening sentences the subspecies that is present in Europe. This is clear later in the manuscript, but it could be mentioned earlier (e.g., lines 66-68)
3. Supplemental Figure 2 - please explain the red and blue coloring of the aa residues in the aligments.
Author Response
We would like to thank the reviewer for his helpful comments. Please find below the specific comments and our response in bold
1. The authors state that Aedes vexans is an invasive species, and I'm not sure if there are data to support that. It is certainly widespread ("cosmopolitan") species and locally quite abundant, but I am not sure whether it could be classified as invasive.
We agree with reviewer that there is not sufficient evidence to support the term "invasive" and therefore removed it from the text in line 64
2. Please mention in the opening sentences the subspecies that is present in Europe. This is clear later in the manuscript, but it could be mentioned earlier (e.g., lines 66-68)
We added Europe to line 66
3. Supplemental Figure 2 - please explain the red and blue colouring of the aa residues in the alignments.
We apologise for the oversight, figure legends are now changed to reflect that “sequence conservation is indicated by colouring 0% (blue) 50% (black) and 100% (red).” Additionally, we have added a legend in order to make the alignment visually clear. This is now represented in Figure S3.
Reviewer 3 Report
This is a well-written manuscript describing identification of a novel iflavirus in Aedes vexans. The authors mined previously published sRNA data and then assembled the putative iflavirus genome. They describe overall coverage ranging from ~9->2000. They describe the predicted virus structural features and their similarity to other related genomes. Then they re-examined the sRNA reads in the context of the sRNA response to iflavirus infection.
Major issues
Because the metagenome was manually stitched together and not validated by de novo full length sequencing or PCR, how can you be sure that the gene order is correct? The gene order of published iflaviruses varies, and the proposed gene order is different than that of another recently identified mosquito iflavirus that was sequenced using alternative sequencing methods (Cholletti et al., Virol J. 2018; 15: 71).
Please provide some support for your decision to manually stitch together the viral genome. Why wasn’t SPAdes suitable for obtaining a complete assembly?
Can you show reproducibility of the genome orientation using PCR or some other method?
Ln 406- The authors state that mosquito iflaviruses show little sequence similarity. Did you try to align the SPAdes contigs to those viruses? What was the sequence similarity?
Specifically, in what ways does the Aedes vexans viral sequence differ from previously identified mosquito iflaviruses?
Minor-
Ln 152, correct to “SPAdes”
Ln 171 Why did you choose the invertebrate mitochondrial genome for the genetic code in ORF finder?
Ln 203- what were the nt positions of the RT-PCR primers?
Ln 242 Aedes vexans should be italicized
Ln 301 Please provide a citation for “Generally, iflaviruses do not appear to cluster together with related insect hosts”. The lack of within genera hosts could be merely due to the lack of identification rather than extreme divergence of these insect-only viruses.
Ln 450 Define RNASEK.
Author Response
We appreciate the very insightful comments of the reviewer and we have added some new analysis and a new supplementary figure to assure that the novel genome we present has been validated appropriately. The concerns raised by the reviewer are listed blow and our answers are given in bold.
Because the metagenome was manually stitched together and not validated by de novo full length sequencing or PCR, how can you be sure that the gene order is correct? The gene order of published iflaviruses varies, and the proposed gene order is different than that of another recently identified mosquito iflavirus that was sequenced using alternative sequencing methods (Cholletti et al., Virol J. 2018; 15: 71). Please provide some support for your decision to manually stitch together the viral genome. Why wasn’t SPAdes suitable for obtaining a complete assembly?
We agree that the quality of our silico assembly is key to announce the new viral genome. We apologize for not presenting enough description of the results that would provide confidence in the assembly. We have solved this shortcoming by adding more description to the manuscript (please see corrected manuscript with highlighted changes) and by adding new analysis to confirm our alignments (see Supplementary figure S1). Firstly, while the coverage of the genome is very high with 2 million mapped reads (2,021,588/174,375,127 mapped reads) it is no surprise that genome assembly is computationally challenging given the variable read length and coverage (making a suitable single k-mer difficult). In the interests of reproducibility we have re-downloaded and re-assembled the virus contig using a command line version of SPAdes (SPAdes 3.11.1) with the –rna flag (as this option is not available in the Galaxy front end). We have also used two additional assemblers (CLC Genomics Workbench 12.0.3 and Trinity v2.1.1). In doing so we were able to recover much higher virus contig lengths from SPAdes assembler using the –rna flag (top size 8590nt) and Trinity (6980nt), with CLC Genomics workbench performing not as well (top size 2717nt). To visualize the construction of the genome as well as harmony between assemblers we provide now an additional Supplementary Figure 1.
Based on the orientation of the overlapping portions of the contigs it is impossible to place the overlapping regions to recreate the gene order observed in (Cholleti et al., Virol J. 2018; 15: 71). Additionally, the 6980nt contig from the Trinity assembly covers the whole Leader and non-structural gene order, which are the genes that vary. Given that the reported virus from Cholleti et al. is divergent to Aedes vexans iflavirus and two assemblers have completely assembled the potentially divergent gene order, the genome orientation reported by Cholleti does now impair our result here.
Can you show reproducibility of the genome orientation using PCR or some other method?
We understand that reviewer 3 is concerned that the genes of AvIFV might be orientated based on a report of another mosquito Iflavirus genome (Cholleti et al., Virol J. 2018; 15: 71. In order to assure both reviewer 3 and readers about the assembly we have provided an additional Supplemental Figure 1. Here, using two additional assemblers and compiling the outputs we have demonstrated the reproducibility of the genome orientation. We were able to recreate the analysis with two other assemblers and also recover higher fragment sizes with SPAdes and show that all fragments have a single overlapping region and single orientation, with no alternative overlapping regions (such as inversions). The combination of adequate coverage at each position, overlapping 100% ends and the ability to produce a single open reading frame suggests it is highly unlikely the genome was misassembled. The harmony of these methods with the result in addition to the reasons outlined earlier are sufficient to and give no indication that further PCR validation is necessary.
Ln 406- The authors state that mosquito iflaviruses show little sequence similarity. Did you try to align the SPAdes contigs to those viruses? What was the sequence similarity?
Specifically, in what ways does the Aedes vexans viral sequence differ from previously identified mosquito iflaviruses?
We appreciate that reviewer 3 may want additional analysis of divergent mosquito iflaviruses, however, that is not the scope of the paper. Beyond describing the genome features and phylogenetic placement we do not believe there is much more to be understood from comparing viruses reported by (Cholleti et al., Virol J. 2018; 15: 71) or other mosquito iflaviruses. We have cited Cholleti et al. in text as reference 55.
Minor-
Ln 152, correct to “SPAdes”
Corrected
Ln 171 Why did you choose the invertebrate mitochondrial genome for the genetic code in ORF finder?
We appreciate that this was a mistake, we have corrected the way we have written this in the text. Reanalysis of this contig removes the two small ORFs that have been predicted but nothing else. We have changed Figure 1 and Figure 3 to remove those ORFs and sentences pertaining to the results.
Ln 203- what were the nt positions of the RT-PCR primers?
We have added that information to the body of text (Line 200) “was established using specific primers amplifying AvIFV gRNA at positions 8713-8983”
Ln 242 Aedes vexans should be italicized
As per naming conventions as set out by the ICTV, viruses which have part or whole of a species name are not to be italicised https://talk.ictvonline.org/information/w/faq/386/how-to-write-virus-and-species-names
Ln 301 Please provide a citation for “Generally, iflaviruses do not appear to cluster together with related insect hosts”. The lack of within genera hosts could be merely due to the lack of identification rather than extreme divergence of these insect-only viruses.
This line was intended as an observation of the result we have changed it to “In the phylogenetic analysis presented here iflaviruses do not cluster together with related insect hosts.”
Ln 450 Define RNASEK.
We have changed this to be Ribonuclease K (AeRNASEK) expression
This manuscript is a resubmission of an earlier submission. The following is a list of the peer review reports and author responses from that submission.
Round 1
Reviewer 1 Report
This manuscript describes the RNAi profile in Senegalese Aedes vexans arabiensis infected by a novel iflavivirus. This description is complicated by the analysis being done on mosquitoes experimentally infected with Rift Valley fever virus (RVFV) MP-12. This type of data is often hard to describe thus careful writing is necessary which is not the case for this manuscript. The source and actual number of samples run is unclear. No comparison of non-RVFV MP-12 infected mosquitoes is evident. The inclusion of MP-12 is not well discussed. The purpose and intent of the study is unclear. The supplemental material was not available on the reviewer webpage. There appears to be some unique and useful information in the study but it is very hard to comprehend. Overall, the manuscript needs major modification so that the reader can follow the scientific logic of the study.
Specific comments:
Ln 112-113: Last line is unclear what mosquitoes this sentence is in reference to – presumably the origin of the colony.
Ln 114: What mosquitoes? Should be originated
Ln 117: control animals? Assume you mean mosquitoes
Ln 267-268: Reference?
Figure 4 is not really needed.
Reviewer 2 Report
Manuscript by Parry et al. describes the discovery of a novel iflavirus from Aedes vexans mosquitoes. Authors identify and assemble the genome through RNAi profiling. They also search this virus by PCR in other colonies of Aedes vexans and do not find the virus. The identification and characterization of the viral sequence seem to be well performed and the nucleotide sequence has been submitted to the GeneBank (MN314969) although it has not still been released. My main concern is related to the search for the iflavirus in other mosquito colonies, that resulted negative for all the samples analyzed. See specific comments below.
Major points
Figure 4. The positive PCR control is a 111 nt synthesised iflavirus DNA based on the sequence data described in this manuscript. To enable direct comparison with mosquito derived samples, it should be included as a positive control RNA samples for the iflavivirus from the mosquito pool/colony used for the identification of the iflavivirus, and not from synthesised RNA. This will confirm that the PCR works in the same conditions of the screened samples. This is of special interest considering that all the samples tested resulted negative. Including this control will rule out the possiblility that the PCR works with in vitro synthesised RNA but not with mosquito-derived samples. I could not find the supporting material for review.
Minor points
Fig. 1B. Label the X-axis as Nucleotide position or similar Line 278. Iflaviridae instead of Iflaviviridae. Fig. 3. Label the X-axes.